# Towards Computing an Optimal Abstraction for Structural Causal Models

**Fabio Massimo Zennaro**[1]          **Paolo Turrini**[1]          **Theodoros Damoulas**[1]

[1]University of Warwick, Coventry, United Kingdom

## Abstract

Working with causal models at different levels of abstraction is an important feature of science. Existing work has already considered the problem of expressing formally the relation of abstraction between causal models. In this paper, we focus on the problem of learning abstractions. We start by defining the learning problem formally in terms of the optimization of a standard measure of consistency. We then point out the limitation of this approach, and we suggest extending the objective function with a term accounting for information loss. We suggest a concrete measure of information loss, and we illustrate its contribution to learning new abstractions.

## 1 INTRODUCTION

Understanding causality is a key challenge for modern artificial intelligence (AI) [Schölkopf et al., 2021]. Structural causal models (SCM) [Pearl, 2009] are well-established tools used in statistics and computer science to describe causal systems and to express causal assumptions in a graphical form. One could, for instance, describe the relation between smoking and cancer in a model with few variables of interest represented as nodes (e.g., environment stress, smoking, and cancer), and with directed arcs denoting the hypothesised causal relationships (see Figure 1a).

An SCM is formulated over a set of *relevant* variables, some of which may not be as important for analysing the problem at hand. In our example, for instance, we may not be interested in explicitly modelling the role of the environment in causing smoking or cancer, but it could be sufficient to have a smaller model incorporating only the latter variables (see Figure 1b). Being able to work with SCMs at different levels of abstraction allows us to adjust to our available computational resources, while still getting meaningful results;

it would also allow us to integrate data that may have been collected with models at different resolutions.

But what is the "right" abstraction of an SCM? Answering this question in a rigorous fashion requires tackling several challenges, among which how to define mathematically a relation of abstraction and how to formalize a notion of consistency among models. Answers to these two problems have recently been proposed in the literature [Rubenstein et al., 2017, Beckers and Halpern, 2019, Rischel, 2020] and, building on these contributions, we can now establish a relationship between SCMs and assess their consistency, as sketched in Figure 1c. Despite the recent progress in the literature, the question still remains of how to order abstractions in terms of gains and losses with respect to the original model and how to compute an optimal abstraction.

**Contribution.** In this paper we address the problem of computing an optimal abstraction for SCMs. We define several concrete subproblems, ranging from the simpler question of finding an abstraction between two fully specified SCMs (Figure 2a) to the harder question of being given a starting model and jointly finding an abstraction and an abstracted model (Figure 2b). We phrase the problem of learning an abstraction as an optimization problem where the objective is to maximize key properties of an abstraction. We start by considering the optimization of a standard measure of consistency; however, this approach could lead to optimal, yet doubtfully useful solutions, such as identities or the collapse of all the variables and outcomes onto a single value (Figure 2c). Therefore, we suggest the introduction of a measure that accounts for the information being lost in an abstraction, and which will be used in combination with consistency. We provide the definition of an optimization problem and we illustrate preliminary results on our motivating example showing how a measure of consistency and a measure of information loss capture different aspects and properties of an abstraction. Throughout the paper, we will illustrate ideas relying on a motivating example, for which we will provide formal details in the appendix.

*Accepted for the Causal Representation Learning workshop at the 38th Conference on Uncertainty in Artificial Intelligence (UAI CRL 2022).*

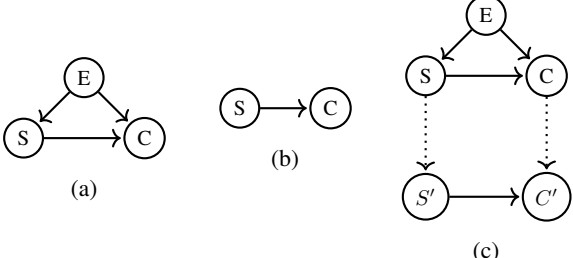

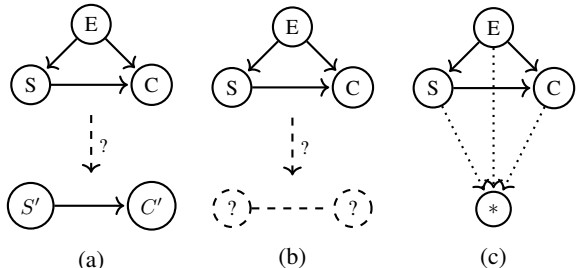

Figure 1: Smoking (S) and cancer (C), at different levels. (a) A simple model, considering the environment (E). (b) Abstracting away from the environment. (c) A sketch of the relationship between the original model and its abstracted version.

**Related Literature.** Abstraction, i.e., the capacity to model a phenomenon with different degrees of detail is ubiquitous in science; in AI, it is fundamental to reduce computational complexity of decision-making and has found important applications in the development of intelligent agents playing complex games at superhuman level [Kroer and Sandholm, 2018].

In the context of SCMs, Rubenstein et al. [2017] proposed to relate causal models via a $(\tau\text{-}\omega)$-*transformation* which connects the space of joint outcomes of all the variables of two SCMs. The requirement of consistency is expressed in terms of interventional consistency: a $(\tau\text{-}\omega)$-transformation is an *exact transformation* if it commutes with respect to a set of interventions of interest. The notion of $(\tau\text{-}\omega)$-transformation has been refined in Beckers and Halpern [2019], Beckers et al. [2020] through stronger definitions meant to rule out degenerate forms of abstractions that would have been admitted under the original definition.

An alternative modelling of abstraction relying on category theory has been proposed by Rischel [2020], Rischel and Weichwald [2021]. Here, an abstraction is defined at two levels: first, as a mapping between the variables of two SCMs; and, second, as mappings between the outcomes of the variables. This setup also admits a way to quantify the degree of approximation or error between two SCMs in case interventional consistency were not to hold. Our work builds over this framework, relying on the rigorous definition of abstraction and the operative definition of abstraction error.

## 2 PRELIMINARIES

### 2.1 STRUCTURAL CAUSAL MODELS

**Definition 1 (SCM [Pearl, 2009])** *A structural causal model (SCM) $\mathcal{M}$ is a tuple $\langle \mathcal{X}, \mathcal{U}, \mathcal{F}, P(\mathcal{U}) \rangle$ with an underlying directed acyclic graph (DAG) $\mathcal{G}_\mathcal{M}$ where:*

Figure 2: Different abstraction problems. (a) Given two SCMs, learn their relative abstraction. (b) Given an SCM, learn an abstracted model and their relative abstraction. (c) A trivial solution to (b).

- $\mathcal{X}$ *is a finite set of $N$ endogenous random variables $X_i$; each variable $X_i$ is associated with a finite set $\mathcal{M}[X_i] = \{x_1, x_2, ..., x_N\}$ of outcomes; we use the boldface notation $\mathbf{X} \subseteq \mathcal{X}$ for subsets of variables, and $\mathcal{M}[\mathbf{X}] = \prod_{X_i \in \mathbf{X}} \mathcal{M}[X_i]$ for the Cartesian product of the sets of outcomes of the variables in $\mathbf{X}$.*

- $\mathcal{U}$ *is a finite set of $N$ exogenous random variables $U_i$, one for each endogenous variable $X_i$; each variable $U_i$ is associated with a set $\mathcal{M}[U_i] = \{u_1, u_2, ..., u_N\}$ of outcomes.*

- $\mathcal{F}$ *is a finite set of $N$ modular measurable structural functions $f_i$, one for each endogenous variable $X_i$; a structural function $f_i : \mathcal{M}[\mathbf{PA}(X_i)] \times \mathcal{M}[U_i] \to \mathcal{M}[X_i]$, where $\mathbf{PA}(X_i) \subseteq \mathcal{X} \setminus X_i$ defines deterministically the value of the random variable $X_i$ given the values of the variables in the set $\mathbf{PA}(X_i)$ and $U_i$.*

- $P(\mathcal{U})$ *is a joint probability distributions over the exogenous variable $U_i$.*

Endogenous variables represents variables of interest, explained by deterministic mechanisms; exogenous variables capture stochastic factors of variance beyond the control of a modeler. Several common assumptions underlying this definition are explicitly stated in Appendix A.

***Example.*** Let us define a simple toy SCM $\mathcal{M}$ for the lung cancer scenario we introduced earlier:

- $\mathcal{X} = \{E, S, C\}$ is the set of endogenous variables containing three binary variables representing respectively level of stress due to the environment, habit of smoking, and presence of lung cancer;

- $\mathcal{U} = \{U_E, U_S, U_C\}$ is the set of exogenous variables containing three binary variables;

- $\mathcal{F} = \{f_E, f_S, f_C\}$ is the set of structural functions such that $E = f_E(U_E)$, $S = f_S(E, U_S)$, and $C = f_C(E, S, U_C)$;

- $P(\mathcal{U}) = P(U_E, U_S, U_C)$ is the joint probability distribution over the exogenous variables.

Figure 1a can now be given a formal reading as the DAG $\mathcal{G}_\mathcal{M}$ underlying the model we have just defined. Notice that, according to the conventions in the field, the figure shows only the endogenous variables. □

SCMs allow for the rigorous definition of interventions:

**Definition 2 (Intervention)** *Given a SCM $\mathcal{M}$, a set of variable $\mathbf{X} \in \mathcal{X}$ together with an associated set of values $\mathbf{x}$, such that for each $X_i \in \mathbf{X}$ there is a $x_i \in \mathcal{M}[X_i]$, an intervention $\iota : do(\mathbf{X} = \mathbf{x})$ is an operator on a SCM that replaces the structural functions $f_i$ with the constants $x_i$.*

Graphically, an intervention $\iota$ mutilates the original DAG $\mathcal{G}_\mathcal{M}$ by removing all incoming edges in $X_i$ and replacing $f_i$ with $x_i$. Thus, the intervention $\iota$ on the SCM $\mathcal{M}$ induces a new post-interventional model $\mathcal{M}_\iota$.

In a SCM, the probability distributions over the exogenous variables can be pushforwarded over the endogenous variables, thus defining joint probabilities $P_\mathcal{M}(\mathbf{X})$ over $\mathbf{X} \subseteq \mathcal{X}$. Furthermore, the finite dimensionality of the outcome sets of the variables in $\mathcal{M}$ allows us to represent a SCM as a tuple $\langle \mathbb{M}[X], \mathbb{M}[\phi_X] \rangle$, where $\mathbb{M}[X]$ is the set of sets $\mathcal{M}[X_i] \cup \{*\}$ and $\mathbb{M}[\phi_X]$ is the set of mechanisms encoding the conditional distribution of an outcome as a stochastic matrix $\mathcal{M}[\phi_{X_i}]$ representing a stochastic map from $\mathcal{M}[\mathbf{PA}(X_i)]$ to $\mathcal{M}[X_i]$ [Rischel, 2020]. Notice that the collection of sets $\mathbb{M}[X]$ includes the singleton set $\{*\}$ which is necessary to express the mechanisms on endogenous variables $X_i$ that are roots in the DAG $\mathcal{G}_\mathcal{M}$.

*Example.* Let us represent our model $\mathcal{M}$ in terms of sets and stochastic matrices:

- $\mathbb{M}[X]$ is given by the singleton set $\{*\}$ and the three binary sets $\mathcal{M}[E] = \mathcal{M}[S] = \mathcal{M}[C] = \{0, 1\}$;
- $\mathbb{M}[\phi_X]$ is given by the column-stochastic matrices $\mathcal{M}[\phi_E]$ with shape $2 \times 1$, $\mathcal{M}[\phi_S]$ with shape $2 \times 2$, and $\mathcal{M}[\phi_C]$ with shape $2 \times 4$.

The formal definition of $\mathcal{M}$ is available in Appendix D.1. □

## 2.2 ABSTRACTION

**Definition 3 (Abstraction [Rischel, 2020])** *Let $\mathcal{M} = \langle \mathbb{M}[X], \mathbb{M}[\phi_X] \rangle$ and $\mathcal{M}' = \langle \mathbb{M}'[X'], \mathbb{M}'[\phi_{X'}] \rangle$ be two SCMs. An abstraction $\boldsymbol{\alpha}$ from $\mathcal{M}$ to $\mathcal{M}'$ is a tuple $\langle R, a, \alpha_i \rangle$ where:*

- *$R \subseteq \mathcal{X}$ defines a subset of relevant variables in $\mathcal{M}$;*
- *$a : R \to \mathcal{X}'$ is a surjective function mapping relevant variables $R$ in $\mathcal{M}$ to variables in $\mathcal{M}'$;*
- *$\alpha_i : \mathcal{M}[a^{-1}(X_i')] \to \mathcal{M}'[X_i']$ is a collection of surjective functions, one for each variable in $\mathcal{M}'$, map-*

*ping the outcomes of variable(s) $a^{-1}(X_i')$ onto the outcomes of variable $X_i'$.*

An abstraction $\boldsymbol{\alpha}$ defines an (asymmetric) relation from a base or low-level model model $\mathcal{M}$ to an abstracted or high-level model model $\mathcal{M}'$.

*Example.* Let us consider again our toy model $\mathcal{M}$ along with a simplified SCM $\mathcal{M}'$ defined over two binary sets $\mathcal{M}'[S'], \mathcal{M}'[C']$ and two stochastic matrices $\mathcal{M}'[\phi_{S'}], \mathcal{M}'[\phi_{C'}]$. We can now institute an abstraction $\boldsymbol{\alpha}$ from $\mathcal{M}$ to $\mathcal{M}'$ by defining:

- $R = \{S, C\}$, evaluating only nodes $S$ and $C$ in $\mathcal{M}$ as relevant to our abstraction;
- $a : R \to \mathcal{X}'$ mapping $S \mapsto S'$, $C \mapsto C'$ specifying how variables in the two levels are related;
- $\alpha_{S'} : \mathcal{M}[S] \to \mathcal{M}'[S']$ and $\alpha_{C'} : \mathcal{M}[C] \to \mathcal{M}'[C']$.

Figure 1c is an illustration of the abstraction we have just defined. The formal definition of $\mathcal{M}'$ is available in Appendix D.2, while the definition of $\boldsymbol{\alpha}$ is in Appendix D.3. □

The definition of abstraction is paired with a requirement of interventional consistency.

**Definition 4 (Zero-error abstraction [Rischel, 2020])** *An abstraction $\boldsymbol{\alpha}$ from $\mathcal{M}$ to $\mathcal{M}'$ is a zero-error abstraction if, for all disjoint sets $\mathbf{X}', \mathbf{Y}'$ in $\mathcal{X}'$, the following diagram commute:*

$$
\begin{array}{ccc}
\mathcal{M}_\iota[a^{-1}(\mathbf{X}')] & \xrightarrow{\mathcal{M}[\tilde{\phi}_{a^{-1}(\mathbf{Y}')}]} & \mathcal{M}_\iota[a^{-1}(\mathbf{Y}')] \\
\downarrow{\alpha_{\mathbf{X}'}} & & \downarrow{\alpha_{\mathbf{Y}'}} \\
\mathcal{M}'_{\iota'}[\mathbf{X}'] & \xrightarrow{\mathcal{M}'[\tilde{\phi}_{\mathbf{Y}'}]} & \mathcal{M}'_{\iota'}[\mathbf{Y}']
\end{array}
$$

*that is, $\alpha_{\mathbf{Y}'} \circ \mathcal{M}[\tilde{\phi}_{a^{-1}(\mathbf{Y}')}] = \mathcal{M}'[\tilde{\phi}_{\mathbf{Y}'}] \circ \alpha_{\mathbf{X}'}$ for all possible interventions $\iota$ on $a^{-1}(\mathbf{X}')$, where $\mathcal{M}[\tilde{\phi}_{a^{-1}(\mathbf{Y}')}]$ and $\mathcal{M}'[\tilde{\phi}_{\mathbf{Y}'}]$ are the stochastic matrices derived from the SCMs encoding the relevant distribution.*

The interpretation of commutativity is straightforward: an abstraction $\boldsymbol{\alpha}$ is a zero-error abstraction if, for any intervention $\iota$ on $a^{-1}(\mathbf{X}')$, we can obtain the same result in two ways: (i) by abstracting to the high-level post-interventional model and then computing the distribution of interest via a high-level mechanism; or, (ii) by computing a distribution via a low-level mechanism first, and then abstracting to high-level.

**Example.** Let us consider the abstraction $\boldsymbol{\alpha}$ between $\mathcal{M}$ and $\mathcal{M}'$ defined above. Let us consider the following two disjoint subsets in $\mathcal{X}'$: $\mathbf{X}' = \{S'\}$ and $\mathbf{Y}' = \{C'\}$. This implies we will be considering interventions $\iota$ of the form $do(S = s)$. To evaluate commutativity we then consider the following diagram:

$$
\begin{array}{ccc}
\mathcal{M}_\iota[S] & \xrightarrow{\left[\begin{smallmatrix} 0.88 & 0.38 \\ 0.12 & 0.62 \end{smallmatrix}\right]} & \mathcal{M}_\iota[C] \\
{\scriptstyle\left[\begin{smallmatrix} 1 & 0 \\ 0 & 1 \end{smallmatrix}\right]}\Big\downarrow & & \Big\downarrow{\scriptstyle\left[\begin{smallmatrix} 1 & 0 \\ 0 & 1 \end{smallmatrix}\right]} \\
\mathcal{M}'_{\iota'}[S'] & \xrightarrow{\left[\begin{smallmatrix} 0.88 & 0.38 \\ 0.12 & 0.62 \end{smallmatrix}\right]} & \mathcal{M}'_{\iota'}[C']
\end{array}
$$

It is immediate to see that the diagram commute. A detailed explanation of the diagram is in Appendix D.4. $\square$

**Non-commutativity.** In case an abstraction diagram were not to commute, we could quantify the discrepancy between the upper and the lower path using Jensen-Shannon distance (JSD) [Cover, 1999] as:

$$
E_{\boldsymbol{\alpha}}(\mathbf{X}', \mathbf{Y}') = \max_\iota D_{JSD}(\alpha_{\mathbf{Y}'} \circ \mathcal{M}[\tilde{\phi}_{a^{-1}(\mathbf{Y}')}], \tag{1}
$$
$$
\mathcal{M}'[\tilde{\phi}_{\mathbf{Y}'}] \circ \alpha_{\mathbf{X}'}).
$$

over all interventions $\iota$ on $a^{-1}(\mathbf{X}')$. A definition of JSD is given in Appendix B.

The choice of using JSD was proposed in Rischel [2020], and justified on the ground that, when composing abstractions, JSD guarantees that the overall error is bounded by the sum of the component errors [Rischel, 2020, Rischel and Weichwald, 2021]. From this measure of error on a single diagram, it is possible to define an overall abstraction error as follows.

**Definition 5 (Abstraction Error [Rischel, 2020])** *Let $\boldsymbol{\alpha}$ be an abstraction from a model $\mathcal{M}$ to a model $\mathcal{M}'$. Then the abstraction error is:*

$$
e(\boldsymbol{\alpha}) = \sup_{\mathbf{X}', \mathbf{Y}' \subseteq \mathcal{X}'} E_{\boldsymbol{\alpha}}(\mathbf{X}', \mathbf{Y}') \tag{2}
$$

*for all disjoint non-empty and non-independent subsets $\mathbf{X}', \mathbf{Y}' \subseteq \mathcal{X}'$.*

**Example.** Let us consider the same base model $\mathcal{M}$ and suppose we are given an alternative abstracted model $\mathcal{M}''$, identical to $\mathcal{M}'$ except for the mechanism $\mathcal{M}''[\phi_{C''}]$ which is now encoded by the matrix $\left[\begin{smallmatrix} 0.8 & 0.3 \\ 0.2 & 0.7 \end{smallmatrix}\right]$. Let us relate $\mathcal{M}$ and $\mathcal{M}''$ via the previous abstraction $\boldsymbol{\alpha}$. We then obtain $E_{\boldsymbol{\alpha}}(S'', C'') \approx 0.077$. Moreover, since the $S'', C''$ are the only two disjoint subsets in $\mathcal{M}''$, we also get that the overall abstraction error $e(\boldsymbol{\alpha}) \approx 0.077$. The formal definition of $\mathcal{M}''$ is available in Appendix D.5, the computation of the abstraction error in Appendix D.6. $\square$

## 3 LEARNING ABSTRACTIONS

The definition of abstraction error provides us with a rigorous way to analyze the quality of an abstraction. We could then consider expressing the problem of learning new abstractions (or improving on existing ones) by defining the optimization problem:

$$
\min_{\boldsymbol{\alpha}} \ e(\boldsymbol{\alpha}) \tag{3}
$$

over the space of abstractions $\boldsymbol{\alpha} = \langle R, a, \alpha \rangle$, and, implicitly, over the space of SCMs $\mathcal{M}'$ implied by such an abstraction.

**Hierarchy of problem.** If we make the optimization variables in Equation 3 explicit, we obtain:

$$
\begin{aligned}
\min_{\substack{ |\mathcal{X}'| \in \mathbb{N} \\ |\mathcal{M}'[X_i']| \in \mathbb{N} \\ \mathcal{M}'[\phi_{X_i'}] \in \mathbb{S}(|\mathcal{M}'[X_i']|, |\mathcal{M}'[\mathbf{PA}(X_i')]|) \\ R \subseteq \mathcal{X} \\ a \in \mathbb{S}_{\{0,1\}}(|\mathcal{X}'|, |\mathcal{X}|) \\ \alpha_{X_i'} \in \mathbb{S}_{\{0,1\}}(|\mathcal{M}'[X_i']|, |\mathcal{M}[a^{-1}(X_i')]|) }} & e(\boldsymbol{\alpha})
\end{aligned} \tag{4}
$$

under the constraints:

$$
\begin{aligned}
\text{s.t. } & \mathcal{G}_{\mathcal{M}'} \text{ is acyclic} \\
& a\mathbf{1}_{|\mathcal{X}|}^T > 1 \\
& \alpha_{X_i'}\mathbf{1}_{|\mathcal{M}[a^{-1}(X_i')]|}^T > 1
\end{aligned}
$$

where $\mathbb{S}(a, b)$ is the space of column-stochastic matrices with dimension $a \times b$, $\mathbb{S}_{\{0,1\}}(a, b)$ is the space of binary column-stochastic matrices with dimension $a \times b$, $\mathbf{1}_k^T$ is a column vector of length $k$ of ones.

Notice how the first three lines of optimization variables account for the learning of model $\mathcal{M}'$, while the last three lines account for the learning of the abstraction $\boldsymbol{\alpha}$; moreover, the first constraint enforces acyclicity, while the last two constraints enforce surjectivity. We can identify different classes of problems according to the variables that are given, as summarized in Table 1 in Appendix C.

The problem in Equation 4 is defined over integer domains, and constitutes a combinatorial optimization problem. We will leave the discussion of its complexity and of efficient algorithms to future work; in our motivating example, given its limited size, we are able to find solutions by enumeration.

**Loss function.** The objective in the optimization problem of Equation 3 might be insufficient for learning a meaningful abstraction. In a problem where we can learn the abstraction $\boldsymbol{\alpha}$ and the abstracted model $\mathcal{M}'$ (like in Figure 2b), a trivial optimal solution would be to learn an abstraction $\boldsymbol{\alpha}$ that maps everything to a singleton SCM (as in Figure 2c). By mapping all the variables in $\mathcal{M}$ to a single variable,

and mapping all possible outcomes of the variables $\mathcal{M}[X_i]$ to a single value, commutativity is trivially preserved and $e(\boldsymbol{\alpha}) = 0$. This is due to the fact that a zero abstraction error only guarantees that by commuting abstraction and mechanisms we will obtain the same result, but it does not take into account the amount of information that is given up in an abstraction.

We then suggest rewriting the objective function as:

$$\min_{\boldsymbol{\alpha}} e(\boldsymbol{\alpha}) + \lambda i(\boldsymbol{\alpha}) \qquad (5)$$

where $i(\boldsymbol{\alpha})$ is a measure of information loss due to the abstraction, and $\lambda \in \mathbb{R}$ is a trade-off parameter.

**Measure of information loss.** Different measure of information loss may be considered; customized measures may weigh the information loss proportionally to the importance of different subsystems, emphasizing the contribution of specific (observational or interventional) conditional distributions.

Here we propose a simple generic measure based on the discrepancy between the observational joint distribution $P_{\mathcal{M}}(\mathcal{X})$ of the low-level model $\mathcal{M}$ and the observational joint distribution $\hat{P}_{\mathcal{M}}(\mathcal{X})$ that we would reconstruct inverting the abstraction $\boldsymbol{\alpha}$. We define the inverse of a function $\alpha_{X'_i}$ as:

$$\alpha^*_{X'_i} = \ell_{1,col}(\alpha^T_{X'_i}) \qquad (6)$$

where $\ell_{1,col}$ is an $\ell_1$-normalization along the columns, and $\cdot^T$ is the transpose operator. Although for binary column-stochastic matrices this inverse is just the conventional Moore-Penrose pseudoinverse, the formulation in Equation 6 highlights the rationale behind this choice. By using a transpose, we require to map back a high-level outcome to a low-level outcome; however, multiple low-level outcome may be mapped to a single high-level outcome; by using a $\ell_1$-normalization we evenly spread the probability among all possible low-level outcomes, in accordance with Laplace's principle of insufficient reason [Jaynes, 1957].

If $R = \mathcal{X}_{\mathcal{M}}$, we can then define a global inverse as:

$$\alpha^* = \bigotimes_{X'_i \in \mathcal{X}_{\mathcal{M}'}} \alpha^*_{X'_i}, \qquad (7)$$

where $\otimes$ is the Kronecker product. If $R \subset \mathcal{X}_{\mathcal{M}}$, we need to account for non-relevant variables. Let $\bar{R}$ be the set of non-relevant variables, and let $r$ be the cardinality $|\mathcal{M}[\bar{R}]|$; then we can compute the global inverse as:

$$\alpha^* = \frac{\mathbf{1}_r^T}{r} \otimes \left( \bigotimes_{X'_i \in \mathcal{X}_{\mathcal{M}'}} \alpha^*_{X'_i} \right) \qquad (8)$$

where $\mathbf{1}_r^T$ is a column vector of length $r$ of ones.

***Example.*** Let us consider our motivating example for the abstraction $\boldsymbol{\alpha}$ from $\mathcal{M}$ to $\mathcal{M}'$, and compute the global inverse $\alpha^*$:

$$\alpha^* = \left[ \begin{array}{c} .5 \\ .5 \end{array} \right] \otimes \left[ \begin{array}{cc} 1 & 0 \\ 0 & 1 \end{array} \right] \otimes \left[ \begin{array}{cc} 1 & 0 \\ 0 & 1 \end{array} \right]$$

The explicit computation of $\alpha^*$ is given in Appendix D.7. □

Finally, in analogy with abstraction error, we can define our information loss measure as:

$$i(\boldsymbol{\alpha}) = D_{JSD}(P_{\mathcal{M}}(\mathcal{X}), \alpha^*(P_{\mathcal{M'}})(\mathcal{X}))). \qquad (9)$$

***Example.*** The information loss for our motivating example is:

$$i(\boldsymbol{\alpha}) \approx 0.44.$$

Exact computations are reported in Appendix D.8. □

Information loss provides a different criterion for evaluating abstraction, and it allows us to quantify two ways in which a base model and an abstracted model may diverge.

**Information loss as discrepancy between distributions.** Interventional consistency is concerned with interventional quantities and mechanisms; it does not take into account marginal distributions on root variables in the DAG of the models, or conditional distributions that do not correspond to any mechanism. Information loss, on the other hand, is measured with respect to the joint distribution of the models and it is sensitive to the values of all the distributions.

Disregarding the value of marginal distributions makes an interventionally-consistent abstraction more robust with respect to shifts in the underlying population which are encoded in probabilities over the root nodes; in an interventional settings this a desirable properties. However, if we were to work in an observational setting, and we were interested in making predictions, especially in the anti-causal direction, a proper reconstruction of the populations may be in order.

***Example.*** The abstraction $\boldsymbol{\alpha}$ from $\mathcal{M}$ to $\mathcal{M}'$ turned out to have zero abstraction error $e(\boldsymbol{\alpha}) = 0$, but quite a high information loss $i(\boldsymbol{\alpha}) \approx 0.44$. This is not surprising if we look at the difference between the marginal distribution over the variable $S$: the two models were likely inferred over populations with almost diametrically opposed smoking patterns. Exact values for the marginals are given in Appendix D.9. This would of course impact observational inferences that we may want to perform on the two models. For instance, if we would like to estimate the (anti-causal) probability that a patient is a smoker, given her cancer status, we could come to different conclusions. See Appendix D.10 for a computation of these conditionals. □

Better abstraction for predictive tasks may then be learned by trying to negotiate interventional consistency and information loss.

*Example.* If we keep our models $\mathcal{M}$ and $\mathcal{M}'$ fixed, it may come to no surprise that an alternative abstraction $\beta$ that swaps the outcomes of the variables in $\mathcal{M}$ and $\mathcal{M}'$ could achieve a lower information loss of $i(\beta) \approx 0.31$, although at the cost of not being interventionally consistent anymore, $e(\beta) = 0.22$. Complete definition of $\beta$ is given in Appendix D.11. $\square$

*Example.* If in our optimization we can learn a different abstracted model, it would be possible to suggest an alternative $\mathcal{M}'''$ with a distribution on $\mathcal{M}'''[S''']$ that, while retaining interventional consistency, reduces the information loss of the abstraction $\alpha$ to $i(\alpha) \approx 0.24$. Complete definition of $\mathcal{M}'''$ is given in Appendix D.13. $\square$

**Information loss as quantification of uncertainty due to reduction in resolution.** Furthermore, information loss may act as a proxy to quantify how much detail is lost through abstraction. Reducing the number of variables in an abstracted model, or restricting the range of outcomes of the same variables, implies more uncertainty in the reconstruction of the joint distribution over the base model via the inverse $\alpha^*$. Notice, however, that information loss is a function of the reconstructed probability, not of the number of variables or their cardinality; if the base joint distribution over a set of variables is already maximally uncertain, there will be no information loss in coarsening these variables together. Abstracting a model to a singleton like in Figure 2c would then be sensible when the uncertainty of the base model is so high that we would not lose much by working on the abstracted singleton model and then reconstructing the original distribution.

*Example.* Let us consider the original base low-level model $\mathcal{M}$, and let us instantiate a singleton model $\mathcal{M}^s$, together with the trivial abstraction $\gamma$ from $\mathcal{M}$ to $\mathcal{M}^s$. This abstraction has zero abstraction error $e(\gamma) = 0$, and an information loss of $i(\gamma) \approx 0.37$. Notice that the information loss $i(\gamma)$ is less that the information loss for the abstraction $\alpha$ from $\mathcal{M}$ to $\mathcal{M}'$, despite $\mathcal{M}'$ being defined on more variables; this is due to the fact that the joint distribution reconstructed via the inverse $\alpha^*$ is further from the original joint distribution than the maximally uncertain distribution reconstructed via the inverse $\gamma^*$. However, information loss $i(\gamma)$ is higher than the information loss for the abstraction $\alpha$ from $\mathcal{M}$ to $\mathcal{M}'''$; in this case, the mapping to $\mathcal{M}'''$ successfully exploits the higher number of variables and their cardinality to retain statistical information from the base model. The exact definition of the singleton model $\mathcal{M}^s$ is available in Appendix D.14, the abstraction $\gamma$ in Appendix D.15. $\square$

# 4 DISCUSSION

In this paper, we considered the problem of learning abstractions between SCMs: we introduced a taxonomy of optimization problems, we highlighted the limitation of focusing only on consistency, we proposed a tentative definition of an information loss quantity, and we illustrated the relevance of such a measure. Future work will take into account evaluating the complexity of the identified problem, justifying a proper information loss measure, evaluating its properties and trade-offs, and proposing heuristics for the learning problem.

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

# A SCM ASSUMPTIONS

In our definition of SCM we make the following assumptions:

1. *(Finite Variables)* We explicitly assumed that a SCM is defined on a finite number of $N$ endogenous variables.

2. *(Unique Exogenous Variable (UEV))* With no loss of generality [Beckers and Halpern, 2019], we assumed a single exogenous variable associated with each endogenous variable.

3. *(Non-independent Exogenous Variables)* We do not assume exogenous variables to be independent. This allows the exogenous variables to define a latent structure.

4. *(Modularity)* The mechanisms encoded by the structural functions are independent of each other. This assumption is necessary to specify perfect interventions.

5. *(Measurability)* The structural functions encoding the mechanisms are measurable functions. This assumption is necessary to guarantee that we can pushforward the probability distributions over the exogenous variables onto the endogenous variables.

6. *(Acyclicity)* Every SCM $\mathcal{M}$ admits an underlying graph $\mathcal{G}_{\mathcal{M}} = \langle V, E \rangle$, where $V = \mathcal{X} \cup \mathcal{U}$ is the set of vertices given by endogenous and exogeonous variables, and $E$ is the set of edges determined by the structural functions in $\mathcal{F}$; precisely, for each structural function $f_i$, and for each variable $Y \in \mathbf{PA}(X_i) \cup \{U_i\}$, we will introduce and edge from $Y$ to $X_i$. Notice that, under this construction, the set $\mathbf{PA}(X_i)$ can be given the graph-theoretic reading of *parents of $X_i$*. We assume that the graph $\mathcal{G}_{\mathcal{M}}$ is acyclic.

7. *(Finite Domains)* Following Rischel [2020] we will assume that the domain of each endogenous variable $\mathcal{M}[X_i] = \{x_1, x_2, ..., x_{n_i}\}$ is finite. This assumption is necessary to admit a representation of a SCM in terms of sets and stochastic matrices.

Notice that Assumption (3) and (6) imply that our SCM is *semi-Markovian*. For further discussion of these properties see, for instance, Pearl [2009].

# B DEFINITION OF JENSEN-SHANNON DISTANCE

Here we provide a summary definition of *Kullback–Leibler divergence* and *Jensen-Shannon distance* between two discrete probability distributions. For a more generic treatment of probability distances and their properties, we refer the reader to Cover [1999].

**Definition 6 (Kullback–Leibler (KL) divergence)** *Let $p$ and $q$ be two probability mass functions defined on the same domain $\mathcal{X}$, such that $p(x) > 0$ for all $x \in \mathcal{X}$. The Kullback–Leibler (KL) divergence from $p$ to $q$ is defined as:*

$$d_{KL}(p; q) = \sum_{x \in \mathcal{X}} p(x) \log \frac{q(x)}{p(x)}.$$

**Definition 7 (Jensen-Shannon (JSD) distance)** *Let $p$ and $q$ be two probability mass functions defined on the same domain $\mathcal{X}$, such that $p(x) > 0$ and $q(x) > 0$ for all $x \in \mathcal{X}$. The Jensen-Shannon distance between $p$ and $q$ is defined as:*

$$D_{JSD}(p, q) = \frac{1}{2} d_{KL}(p; m) + \frac{1}{2} d_{KL}(q; m),$$

*where $m = \frac{1}{2}p + \frac{1}{2}q$.*

# C HIERARCHY OF LEARNING PROBLEMS

Table 1 reports a listing of relevant abstraction learning problems.

# D MOTIVATING EXAMPLE

Here is a full specification of the motivating example we used throughout the paper. Code for these models is available at `https://github.com/FMZennaro/CategoricalCausalAbstraction/blob/main/P1%20-%20Motivating%20Example.ipynb`.

## D.1 LOW-LEVEL MODEL $\mathcal{M}$

Let our low-level model $\mathcal{M}$ be defined by

- $\mathbb{M}[X]$ containing the following sets:
  - $\{*\}$
  - $\mathcal{M}[E] = \{0, 1\}$
  - $\mathcal{M}[S] = \{0, 1\}$
  - $\mathcal{M}[C] = \{0, 1\}$
- $\mathbb{M}[\phi_X]$ given by the following column-stochastic matrices:
  - $\mathcal{M}[\phi_E]$: stochastic map from $\{*\}$ to $\mathcal{M}[E]$ encoded by matrix $\begin{bmatrix} 0.8 \\ 0.2 \end{bmatrix}$ representing $P_{\mathcal{M}}(E)$;
  - $\mathcal{M}[\phi_S]$: stochastic map from $\mathcal{M}[E]$ to $\mathcal{M}[S]$ encoded by matrix $\begin{bmatrix} 0.8 & 0.6 \\ 0.2 & 0.4 \end{bmatrix}$ representing $P_{\mathcal{M}}(S|E)$;
  - $\mathcal{M}[\phi_C]$: stochastic map from $\mathcal{M}[E] \times \mathcal{M}[S]$ to $\mathcal{M}[C]$ encoded by matrix $\begin{bmatrix} 0.9 & 0.8 & 0.4 & 0.3 \\ 0.2 & 0.4 & 0.6 & 0.7 \end{bmatrix}$ representing $P_{\mathcal{M}}(C|E, S)$.

| $\mathcal{M}$ | $\mathcal{M}'$ | | | Abstraction | | | Problem |
|---|---|---|---|---|---|---|---|
| | $|\mathcal{X}'|$ | $|\mathcal{M}'[X'_i]|$ | $\mathcal{M}'[\phi_{X'_i}]$ | $R$ | $a$ | $\alpha_{X'_i}$ | |
| given | given | given | given | given | given | given | *Assessment problem:* everything is fully specified. We want to check the degree of consistency and information loss. |
| given | given | given | given | given | given | - | *Completion/fixing problem:* everything is specified except for some or all mappings between outcomes. We want to design or fix the binary stochastic matrices ($\alpha_{X'_i}$) that minimize a loss. |
| given | given | given | given | - | - | - | *Abstraction design problem:* only the models are given. We want to decide how to map low-level variables to high-level variables ($R$ and $a$) and design the binary stochastic matrices ($\alpha_{X'_i}$) that minimize a loss. |
| given | given | given | - | - | - | - | *Abstraction and mechanism design problem:* the base model is completely specified, while for the abstracted model we only have the variables and their domains. We want to find high-level mechanisms ($\mathcal{M}'[\phi_{X'_i}]$), how to map low-level variables to high-level variables ($R$ and $a$) and design the binary stochastic matrices ($\alpha_{X'_i}$) that minimize a loss. |
| given | given | - | - | - | - | - | *Abstraction and granularity design problem:* the base model is completely specified, while for the abstracted models we only know the variables it is defined over, but not their domain or their mechanisms. We want to decide the cardinality of the domain of the high-level variables ($|\mathcal{M}'[X'_i]|$), find high-level mechanisms ($\mathcal{M}'[\phi_{X'_i}]$), how to map low-level variables to high-level variables ($R$ and $a$) and design the stochastic-binary matrices ($\alpha_{X'_i}$) that minimize a loss. |
| given | - | - | - | - | - | - | *Abstracted model design problem:* we are only given the base model. We want to design an abstracted model and an abstraction in all their details so that they minimize a loss. |
| - | given | given | given | - | - | - | *Inverse abstracted model design problem* |

Table 1: Hierarchy of abstraction learning problems.

## D.2 HIGH-LEVEL MODEL $\mathcal{M}'$

Let the high-level model $\mathcal{M}'$ be defined by:

- $\mathbb{M}'[X']$ containing the following sets:
  - $\{*\}$
  - $\mathcal{M}'[S'] = \{0,1\}$
  - $\mathcal{M}'[C'] = \{0,1\}$
- $\mathbb{M}'[\phi_{X'}]$ given by the following column-stochastic matrices:
  - $\mathcal{M}'[\phi_{S'}]$: stochastic map from $\{*\}$ to $\mathcal{M}'[S']$ encoded by matrix $\begin{bmatrix} 0.2 \\ 0.8 \end{bmatrix}$ representing $P_{\mathcal{M}'}(S')$;
  - $\mathcal{M}'[\phi_{C'}]$: stochastic map from $\mathcal{M}'[S']$ to $\mathcal{M}'[C']$ encoded by matrix $\begin{bmatrix} 0.88 & 0.38 \\ 0.12 & 0.62 \end{bmatrix}$ representing $P_{\mathcal{M}'}(C'|S')$.

## D.3 ABSTRACTION $\alpha$

Let the abstraction $\alpha$ from $\mathcal{M}$ to $\mathcal{M}'$ be defined by

- $R = \{S, C\}$;
- $a : R \to \mathcal{X}'$ mapping $S \mapsto S', C \mapsto C'$;
- $\alpha$ given by the collection of maps:
  - $\alpha_{S'} : \mathcal{M}[S] \to \mathcal{M}'[S']$ encoded by matrix $\begin{bmatrix} 1 & 0 \\ 0 & 1 \end{bmatrix}$;
  - $\alpha_{C'} : \mathcal{M}[C] \to \mathcal{M}'[C']$ encoded by matrix $\begin{bmatrix} 1 & 0 \\ 0 & 1 \end{bmatrix}$.

## D.4 COMMUTING DIAGRAM FOR ABSTRACTION $\alpha$ WHEN CONSIDERING SETS $S'$ AND $C'$

Let us evaluate the commutativity of abstraction $\alpha$ from $\mathcal{M}$ to $\mathcal{M}'$ when considering the disjoint sets $S'$ and $C'$. We consider the following diagram:

where:

- the left vertical arrow $\begin{bmatrix} 1 & 0 \\ 0 & 1 \end{bmatrix}$ encodes the abstraction $\alpha_{S'}$ mapping deterministically values of $S$ to values of $S'$;

- the upper horizontal arrow $\begin{bmatrix} 0.88 & 0.38 \\ 0.12 & 0.62 \end{bmatrix}$ encodes a (virtual) mechanism $\mathcal{M}[\tilde{\phi}_C]$ from $S$ to $C$; notice that this mechanism is computed as $P_{\mathcal{M}}(C|do(S))$ and, as such, it is different from the given mechanism $\mathcal{M}[\phi_C]$ which instead encodes $P_{\mathcal{M}}(C|E,S)$;

- the right vertical arrow $\begin{bmatrix} 1 & 0 \\ 0 & 1 \end{bmatrix}$ encodes the abstraction $\alpha_{C'}$ mapping deterministically values of $C$ to values of $C'$;

- the lower horizontal arrow $\begin{bmatrix} 0.88 & 0.38 \\ 0.12 & 0.62 \end{bmatrix}$ encodes a (virtual) mechanism $\mathcal{M}'[\tilde{\phi}_{C'}]$ from $S'$ to $C'$; this mechanism is computed as $P_{\mathcal{M}'}(C'|do(S'))$ and, in this case, it is the same as the given mechanism $\mathcal{M}'[\phi_{C'}]$ which encodes $P_{\mathcal{M}'}(C'|S')$.

## D.5 HIGH-LEVEL MODEL $\mathcal{M}''$

Let an alternative high-level model $\mathcal{M}''$ be defined by:

- $\mathbb{M}''[X'']$ containing the following sets:
  - $\{*\}$
  - $\mathcal{M}''[S''] = \{0,1\}$
  - $\mathcal{M}''[C''] = \{0,1\}$
- $\mathbb{M}''[\phi_{X''}]$ given by the following column-stochastic matrices:
  - $\mathcal{M}''[\phi_{S''}]$: stochastic map from $\{*\}$ to $\mathcal{M}''[S'']$ encoded by matrix $\begin{bmatrix} 0.2 \\ 0.8 \end{bmatrix}$ representing $P_{\mathcal{M}''}(S'')$;
  - $\mathcal{M}''[\phi_{C''}]$: stochastic map from $\mathcal{M}''[S'']$ to $\mathcal{M}''[C'']$ encoded by matrix $\begin{bmatrix} 0.8 & 0.3 \\ 0.2 & 0.7 \end{bmatrix}$ representing $P_{\mathcal{M}''}(C''|S'')$.

## D.6 ABSTRACTION ERROR $e(\alpha)$ FOR $\alpha$ FROM $\mathcal{M}$ TO $\mathcal{M}''$

Let us consider the two disjoint subsets in $\mathcal{X}''$: $\{S''\}$ and $\{C''\}$. To evaluate the abstraction error $E_\alpha(S'', C'')$ of the abstraction $\alpha$ from $\mathcal{M}$ to $\mathcal{M}''$ we consider the following diagram:

and we evaluate:

$$E_{\boldsymbol{\alpha}}(S'', C'') = \max\{D_{JSD}([0.88, 0.12], [0.8, 0.2]),$$
$$D_{JSD}([0.38, 0.62], [0.3, 0.7])\}$$
$$\approx 0.077.$$

### D.7 INVERSE $\alpha^*$

Let us compute the global inverse $\alpha^*$:

$$\alpha^* = \frac{1}{2}\begin{bmatrix} 1 \\ 1 \end{bmatrix} \otimes \alpha^*_{S'} \otimes \alpha^*_{C'}.$$

The inverses $\alpha^*_{S'}$ and $\alpha^*_{C'}$ are trivially identities. The global inverse $\alpha^*$ is then given by:

$$\alpha^* = \begin{bmatrix} .5 \\ .5 \end{bmatrix} \otimes \begin{bmatrix} 1 & 0 \\ 0 & 1 \end{bmatrix} \otimes \begin{bmatrix} 1 & 0 \\ 0 & 1 \end{bmatrix}$$

$$= \begin{bmatrix} .5 & & & & & & & \\ & .5 & & & & & & \\ & & .5 & & & & & \\ & & & .5 & & & & \\ .5 & & & & & & & \\ & .5 & & & & & & \\ & & .5 & & & & & \\ & & & .5 & & & & \end{bmatrix}$$

where, for readability, we omitted writing zeros in the last matrix. Notice how the matrix $\alpha^*$ expresses our uncertainty in reconstructing $\mathcal{M}$ from $\mathcal{M}'$: for instance, the first column of the matrix $\alpha^*$ encodes the fact that the joint values $(S' = 0, C' = 0)$ could be evenly mapped to the joint values $(E = 0, S = 0, C = 0)$ or $(E = 1, S = 0, C = 0)$.

### D.8 INFORMATION LOSS $i(\boldsymbol{\alpha})$ FOR $\alpha$ FROM $\mathcal{M}$ TO $\mathcal{M}'$

In order to compute the information loss, we need to evaluate first the joint distribution on the base model:

$$P_{\mathcal{M}}(E, S, C) = \begin{bmatrix} 0.576 \\ 0.064 \\ 0.064 \\ 0.096 \\ 0.096 \\ 0.024 \\ 0.024 \\ 0.056 \end{bmatrix},$$

then the joint distribution on the abstracted model:

$$P_{\mathcal{M}'}(S', C') = \begin{bmatrix} 0.176 \\ 0.024 \\ 0.304 \\ 0.496 \end{bmatrix},$$

and last reconstruct the distribution over $E, S, C$ via $\alpha^*$:

$$\alpha^*(P_{\mathcal{M}'})(E, S, C) = \begin{bmatrix} 0.088 \\ 0.012 \\ 0.152 \\ 0.248 \\ 0.088 \\ 0.012 \\ 0.152 \\ 0.248 \end{bmatrix}.$$

Finally, we can compute the actual information loss as:

$$D_{JSD}(P_{\mathcal{M}}(E, S, C), \alpha^*(P_{\mathcal{M}'})(E, S, C)) \approx 0.44.$$

### D.9 MARGINAL $P(S)$ IN $\mathcal{M}$ AND $\mathcal{M}'$

Let us evaluate the marginal distribution for the smoking variable $(S)$ in the two models $\mathcal{M}$ and $\mathcal{M}'$. In the base model we have:

$$P_{\mathcal{M}}(S) = \sum_{E,C} P_{\mathcal{M}}(E, S, C) = \begin{bmatrix} 0.76 \\ 0.24 \end{bmatrix}.$$

In the abstracted model we are given:

$$P_{\mathcal{M}'}(S') = \begin{bmatrix} 0.2 \\ 0.8 \end{bmatrix}.$$

The marginal distribution reconstructed via $\alpha^*$ is trivially:

$$\alpha^*(P_{\mathcal{M}})(S) = \sum_{E,C} \alpha^*(P_{\mathcal{M}'})(E, S, C) = \begin{bmatrix} 0.2 \\ 0.8 \end{bmatrix}.$$

### D.10 CONDITIONAL $P(S|C)$ IN $\mathcal{M}$ AND $\mathcal{M}'$

Let us evaluate the conditional distribution of the smoking variable $(S)$ given the cancer variable in the two models $\mathcal{M}$ and $\mathcal{M}'$. In the base model we have:

$$P_{\mathcal{M}}(S|C) = \frac{P_{\mathcal{M}}(S, C)}{P_{\mathcal{M}}(C)} = \begin{bmatrix} 0.88 \\ 0.37 \\ 0.12 \\ 0.63 \end{bmatrix}.$$

In the base model, no-cancer is highly correlated with not-smoking, and having cancer is correlated with smoking. If we were to make this inference in the base model $\mathcal{M}$ and then abstract the outcome via $\alpha_{S'}$, we would map the outcome $S = 0$ to $S' = 0$, and $S = 1$ to $S' = 1$.

However, if we were to abstract the condition via $\alpha_{C'}$, we would first map the condition $C = 0$ to $C' = 0$, and $C = 1$ to $C' = 1$; then if we were to compute the conditional in the abstracted model we would get:

$$P_{\mathcal{M}'}(S'|C') = \frac{P_{\mathcal{M}'}(S', C')}{P_{\mathcal{M}'}(C')} = \begin{bmatrix} 0.37 \\ 0.05 \\ 0.63 \\ 0.95 \end{bmatrix}.$$

Thus, in this case we would infer smoking with high probability for any value of the cancer variable.

## D.11 ABSTRACTION $\beta$

Let the abstraction $\beta$ from $\mathcal{M}$ to $\mathcal{M}'$ be defined by

- $R = \{S, C\}$;
- $b : R \to \mathcal{X}'$ mapping $S \mapsto S'$, $C \mapsto C'$;
- $\beta$ given by the collection of maps:
  - $\beta_{S'} : \mathcal{M}[S] \to \mathcal{M}'[S']$ encoded by matrix $\begin{bmatrix} 0 & 1 \\ 1 & 0 \end{bmatrix}$;
  - $\beta_{C'} : \mathcal{M}[C] \to \mathcal{M}'[C']$ encoded by matrix $\begin{bmatrix} 0 & 1 \\ 1 & 0 \end{bmatrix}$.

## D.12 INVERSE $\beta^*$

The inverses $\beta_{S'}^*$ and $\beta_{C'}^*$ remain exchange matrices. The global inverse $\beta^*$ is then given by:

$$\beta^* = \begin{bmatrix} .5 \\ .5 \end{bmatrix} \otimes \begin{bmatrix} 0 & 1 \\ 1 & 0 \end{bmatrix} \otimes \begin{bmatrix} 0 & 1 \\ 1 & 0 \end{bmatrix}$$

$$= \begin{bmatrix} & & & & & & & .5 \\ & & & & & & .5 & \\ & & & & & .5 & & \\ & & & & .5 & & & \\ & & & .5 & & & & \\ & & .5 & & & & & \\ & .5 & & & & & & \\ .5 & & & & & & & \end{bmatrix}$$

where, for readability, we omitted writing zeros in the last matrix.

## D.13 HIGH-LEVEL MODEL $\mathcal{M}'''$

Let us consider a third high-level model $\mathcal{M}'''$ be defined by:

- $\mathbb{M}'''[X''']$ containing the following sets:
  - $\{*\}$
  - $\mathcal{M}'''[S'''] = \{0, 1\}$
  - $\mathcal{M}'''[C'''] = \{0, 1\}$
- $\mathbb{M}'''[\phi_{X'''}]$ given by the following column-stochastic matrices:
  - $\mathcal{M}'''[\phi_{S'''}]$: stochastic map from $\{*\}$ to $\mathcal{M}'''[S''']$ encoded by matrix $\begin{bmatrix} 0.8 \\ 0.2 \end{bmatrix}$ representing $P_{\mathcal{M}'''}(S''')$
  - $\mathcal{M}'''[\phi_{C'''}]$ : stochastic map from $\mathcal{M}'''[S''']$ to $\mathcal{M}'''[C''']$ encoded by matrix $\begin{bmatrix} 0.88 & 0.38 \\ 0.12 & 0.62 \end{bmatrix}$ representing $P_{\mathcal{M}'''}(C'''|S''')$

## D.14 HIGH-LEVEL MODEL $\mathcal{M}^{\int}$

Let the singleton high-level model $\mathcal{M}^s$ be defined by

- $\mathbb{M}^s[X]$ containing the following set:
  - $\{*\}$
- $\mathbb{M}^s[\phi_*]$ given by the following column-stochastic matrix:
  - $\mathcal{M}^s[\phi_*]$ : stochastic map from $\{*\}$ to $\{*\}$ encoded by matrix $\begin{bmatrix} 1 \end{bmatrix}$

## D.15 ABSTRACTION $\gamma$

Let the abstraction $\gamma$ from $\mathcal{M}$ to $\mathcal{M}^s$ be defined by:

- $R = \{E, S, C\}$;
- $c : R \to \mathcal{X}^s$ mapping $E \mapsto \{*\}$, $S \mapsto \{*\}$, $C \mapsto \{*\}$;
- $\gamma$ given by the map:
  - $\gamma_* : \mathcal{M}[S] \times \mathcal{M}[E] \times \mathcal{M}[C] \to \{*\}$ encoded by matrix $\begin{bmatrix} 1 & 1 & 1 & 1 & 1 & 1 & 1 & 1 \end{bmatrix}$.

## D.16 INVERSE $\gamma^*$

The global inverse $\gamma^*$ is trivially:

$$\gamma^* = \begin{bmatrix} .125 \\ .125 \\ .125 \\ .125 \\ .125 \\ .125 \\ .125 \\ .125 \end{bmatrix}.$$