# OpenReview forum: "Towards Computing an Optimal Abstraction for Structural Causal Models"
_auai.org/UAI/2022/Workshop/CRL — CRL@UAI 2022 Poster_

### Official Review · Reviewer_hXKj · 2022-06-13

**Rating:** 6
**Confidence:** 4

**Review:**

### Summary of the paper
This work introduces an optimization problem meant to identify interesting causal abstraction. The objective to minimize contains two terms, one measuring interventional inconsistency, and another measuring informational loss. The latter term prevents the learner from finding degenerate solutions.

### Disclaimer
I am not an expert on causal abstraction of SCMs, hence it is difficult for me to assess novelty and whether the literature review was exhaustive enough.

### Review
The idea presented in this work is interesting and I appreciate that every new concept was introduced with an example to make these abstract notions more concrete. However, I some technical statements lacked clarity. I provide examples below. Given that this is a workshop, I will nevertheless recommend acceptance.

Definition of $\mathcal{M}[\phi_{X_i}]$ just before the example of Section 2.1 was confusing. It is introduced as both a map and a stochasitic matrix, but I can't link both. The example afterward clarified, but the confusion came back while parsing the commutating graphs (see next point).

Definition 4: What is $M[\phi_{X_i}]$ and how does it differ from $\mathcal{M}[\phi_{X_i}]$? I will assume that it is a typo, and that they are supposed to be the same. The definition requires that the graph commutes for all disjoint subsets of variables in $\mathcal{X}'$, but the matrix $\mathcal{M}[\phi_{a^{-1}(Y')}]$ is not defined if ${a^{-1}(Y')}$ happens to be a set (with more than one element). Moreover, even if ${a^{-1}(Y')}$ is a singleton, the matrix $\mathcal{M}[\phi_{a^{-1}(Y')}]$ might not take the variables $a^{-1}(X')$ as input. Another point of confusion is that the maps $\mathcal{M}[\phi_{X_i}]$ were supposed to output a distribution over $X_i$, and here the graph suggests that it outputs something in the support of $X_i$. The example above is also confusing. The matrix on the top arrow should have shape 2x4, not 2x2.

Minor:

Appendix A: Assumption 3 allows exogenous variables to be dependent, but the definition of SCM in the main text says that "$\mathcal{P}$ is a set of $N$ probability distributions $p_i$, one for each exogenous variable", which suggests that the endogenous noises are independent (otherwise how does one go from the set of distributions $p_i$ to the joint over the noises?). If you want to allow for dependent noises, you should define a joint distribution over noises, otherwise the SCM is not well defined.

---

### Official Review · Reviewer_DTRq · 2022-06-28
**Interesting but abstract paper on abstraction of causal models**

**Rating:** 6
**Confidence:** 2

**Review:**

This paper builds on some recent literature on abstraction of causal models to propose a notion of an "optimal" abstraction of one model to another. They propose to minimize abstraction error (as defined by Rieschel) subject to a penalty on information loss.

Ultimately it is an interesting beginning to a project that could benefit from more thorough development.

- The specific measure of information loss they introduce seems like one possibly reasonable way to formalize this concept, though it is not fully developed. Its properties are not clear, not clear how to calibrate or interpret different numbers, how it really works in meaningful examples. What alternatives are there and why should this one be preferred?
- How to trade off error and information loss is not clear.
- How would this really be applied outside of toy problems?

---

### Meta-Review · Program_Chairs · 2022-07-06

**Recommendation:** Accept (Poster)
**Confidence:** 4

**Metareview:**

The reviewers agree the paper is sound and should be accepted at the workshop.

---

### Decision · Program_Chairs · 2022-07-06

Accept (Poster)